# Wavefront Sensing for Evaluation of Extreme Ultraviolet Microscopy

**DOI:** 10.3390/s20226426

**Published:** 2020-11-10

**Authors:** Mabel Ruiz-Lopez, Masoud Mehrjoo, Barbara Keitel, Elke Plönjes, Domenico Alj, Guillaume Dovillaire, Lu Li, Philippe Zeitoun

**Affiliations:** 1Deutsches Elektronen-Synchrotron (DESY), 22607 Hamburg, Germany; masoud.mehrjoo@desy.de (M.M.); barbara.keitel@desy.de (B.K.); elke.ploenjes@desy.de (E.P.); 2CNRS, Ecole Polytechique-IPP, ENSTA, Chemin de la Hunière, 91761 Palaiseau, France; domenico.alj@gmail.com (D.A.); philippe.zeitoun@ensta-paris.fr (P.Z.); 3Imagine Optic, 18 Rue Charles de Gaulle, 91400 Orsay, France; gdovillaire@imagine-optic.com; 4Center for Advanced Material Diagnostic Technology, Shenzhen Technology University, Shenzhen 518118, China; lilu@sztu.edu.cn

**Keywords:** wavefront sensor, extreme ultraviolet, optical metrology, Schwarzschild objective

## Abstract

Wavefront analysis is a fast and reliable technique for the alignment and characterization of optics in the visible, but also in the extreme ultraviolet (EUV) and X-ray regions. However, the technique poses a number of challenges when used for optical systems with numerical apertures (NA) > 0.1. A high-numerical-aperture Hartmann wavefront sensor was employed at the free electron laser FLASH for the characterization of a Schwarzschild objective. These are widely used in EUV to achieve very small foci, particularly for photolithography. For this purpose, Schwarzschild objectives require highly precise alignment. The phase measurements acquired with the wavefront sensor were analyzed employing two different methods, namely, the classical calculation of centroid positions and Fourier demodulation. Results from both approaches agree in terms of wavefront maps with negligible degree of discrepancy.

## 1. Introduction

Recent developments in extreme ultraviolet (EUV) lithography have led to a growing interest in high-resolution imaging [1,2,3]. Complex optical systems with two or more mirrors working in the EUV were developed to produce sharp images. In particular, reflective objectives correct third-order aberrations, i.e., spherical, coma, and astigmatism when two curved surfaces are used in compensation. A reduction of higher-order aberrations is possible by adding more curved optical elements in the design of a dioptric system [4], but outcome radiation is drastically reduced as a consequence. The EUV lithography industry has posed a constraint of a numerical aperture (NA) from 0.15 to 0.5 on all-reflective objectives for rendering with resolutions below 30 nm [5,6]. However, to accomplish the desired spatial resolution, this numerical aperture has to be combined with a perfect alignment of the mirrors employed in the system. Therefore, characterization of such optics and minimizing optical aberrations are essential in the development of EUV tools.

The alignment of EUV microscopes such as the Schwarzschild objective, and thus reaching the desired focal-spot quality, is challenging [7,8,9]. Measuring the optical aberrations at-wavelength with a Hartmann wavefront sensor, a well-established technique for EUV wavefront characterization [10,11], is considerably difficult for NA illumination larger than NA > 0.1 due to the following reasons: (i) the phase is strongly curved; (ii) marginal rays have large angles that interact with more than one pinhole of the Hartmann plate, creating nonlinear solutions [12,13]; and (iii) to cover the large divergence of the studied optics, the wavefront sensor has to be placed extremely close to the optics and thus potentially fully inside vacuum. Unfortunately, commercial all-invacuum EUV charge-coupled devices (CCDs) are not very common, while automating movements and implementing water cooling are among potential problems of controlling a camera in vacuum.

For these reasons, Hartmann wavefront sensors working in the EUV regime typically have a numerical aperture of approximately 0.02 [14]. However, Imagine Optic has recently developed a high-numerical-aperture Hartmann wavefront sensor (HNA-HWS) capable of analyzing larger divergence optics [15]. The key of this device is the selection of appropriate parameters for the Hartmann plate: it must be large enough to fully cover the photon beam at a usable distance to the focus with adequate pinhole distribution and proper pinhole diameter [16]. Moreover, the new HNA-HWS is vacuum-compatible, and can therefore be placed very close to the studied optics. This instrument was designed to evaluate geometries up to NA = 0.15. However, it is possible to push the numerical aperture to slightly larger incident angles by adapting the evaluation software to larger and artificial pitch separation. This is needed during evaluation to compensate for optics magnification.

As proof of principle, a Schwarzschild objective is evaluated using a high-numerical-aperture Hartmann wavefront sensor. The aim of our work is to demonstrate that optics with NA > 0.15 can be characterized using wavefront-sensing methods. Two different approaches to solving complicated diffraction patterns and extracting wavefront maps were used: (i) the well-established classical centroid method based on measuring the relative displacements within each subaperture of the grid; and (ii) Fourier demodulation (FD), an approach based on Fourier transformation analysis of measured patterns [17]. Intense EUV radiation is needed to successfully perform such an experiment [13] in single-bunch mode, optimizing data-acquisition time. Our measurements were carried out at the soft X-ray free electron laser FLASH at DESY, Hamburg [18,19,20].

## 2. Materials and Methods

### 2.1. Experiment Setup

The HNA-HWS was tested at the open-port beamline FL21 at FLASH. FLASH was operated at a pulse energy of ∼200 μJ per pulse in single-bunch mode. Wavelength was set at 13.5 ± 0.2 nm (photon energy 90–93 eV). Beam divergence was estimated to be approximately 160 μrad (4-sigma).

The setup for the measurements is described in Figure 1. The photon source was approximately 70 m in front of the experimental vacuum chamber. The Schwarzschild objective (SO) was located in its own vacuum chamber 650 mm downstream of a differential pumping stage (DPS) at pressure of <5 × 10^−6^ mbar. The DPS was made up of two circular apertures with 10 mm diameter separated by 160 mm and the appropriate pumping system. Since the photon-beam diameter at the beamline end was slightly larger than the DPS diameter was, the photon beam was clipped in front of the Schwarzschild chamber. The convex mirror of the Schwarzschild with barely larger diameter (Φ = 10.6 mm) was correctly illuminated, and direct radiation on the sensor was avoided.

### 2.2. Schwarzschild Objective

We characterize a SO designed by Zastrau et al. [21]. The Schwarzschild objective focused approximately 25 millimeters in front of the Hartmann plate, which was protected with a 200 nm thick aluminum filter. In this configuration, additional filters along the beamline were not necessary to decrease intensity. The SO’s architecture was designed in concordance with the Schwarzschild proposed by Artioukov and Krimski in 2000 [22]. Mirrors were coated with high-reflectivity molybdenum and silicon (Mo/Si) multilayers optimized for 13.5 ± 1 nm. Reflectivity was optimized to be higher than 65% for incident angles between 1° and 8°. The total system rendered a transmission of about 45%, and it had a spectral bandwidth of a few percentage points. Full illumination of the mirrors resulted in numerical aperture (NA) = 0.19. The mechanical design was recently improved to reduce the instrument weight and shrink central dark shadowing produced by the convex mirror. Outside of the focus, the Schwarzschild ring shape showed discontinuity at 60°, 180°, and 300°. We observed this effect in the Hartmann pattern shown in Figure 2b,c. The absence of illumination was a consequence of the three fins of the mount that supported the concave mirror.

The mechanical mount was equipped with two picomotors to control the alignment of two angular axes. A third axis, not included in the design, controlled the distance between mirrors (longitudinal alignment). However, for the specific magnification required, the ideal distance between mirrors to satisfy the spherical aberration compensation was the mechanically shortest one (64.5 mm). This distance between mirrors was fixed in the SO before it was mounted in the vacuum chamber.

### 2.3. High-Numerical-Aperture Wavefront Sensor

The HNA-HWS was made up of two components: the so-called Hartmann plate and the CCD camera. The Hartmann panel was a hole plate with 36 mm diameter. Each hole was projected onto the sensor. The hole plate was installed approximately 45 mm from the CCD, and it was manufactured with 50 μm squared holes rotated 22.5° and a grid pitch size of 150 μm [15]. The invacuum water-cooled back-illuminated EUV CCD was sized 28 × 28 mm^2^. This allowed for placing the sensor near the microscope in the same vacuum chamber. The CCD camera had 2048 × 2048 pixels, and pixel size was 13.5 μm.

### 2.4. Analysis Approach

We performed the experiment in two stages. First, pulses of the FEL beam directly illuminated the sensor. Second, the highly focused pulses of the SO were measured. During the direct-beam stage, the FLASH beam was characterized. The phase of an average of 10 pulses was measured to understand the intrinsic phase of the FEL beam ($\phi_{F}$). Wavefronts were reconstructed from the circular aperture of approximately 9 mm. Since the FLASH source was far from the Hartmann sensor, divergence was low enough to obtain adequately distributed local spots at the hole grid. The standard procedure for EUV wavefront measurement followed [23]. The obtained Zernike coefficients were negligible compared to the strongly curved aberrations measured during the second stage, and the wavefront map of the Schwarzschild was evaluated. However, this measurement created more precise interpretation of the aberrations as shown in Equation (Equation 1):(1)$$\phi_{SO}=\phi_{SO+F}-\phi_{F}$$

During this second stage of the experiment, we considered the contributions from the SO mirror misalignments, the FEL beam phase, and the systematic internal misalignment of the sensor. EUV Hartmann wavefront sensors use a reference file for calibration of those internal misalignments [24]. For this aim, a reference wave needs to be acquired. To determine the reference, the image of a well-defined spherical wavefront created by a pinhole is recorded. However, as shown in Figure 2, such a reference file was not usable for our specific measurements because the first SO mirror shadowed the central part of the photon beam, and very strong magnification was created by the optics. Figure 2 shows images of the pinholes of the Hartmann plate for different cases: (a) for a direct beam without magnification, (b) after propagating through the Schwarzschild with a distance between the SO focus and the Hartmann grid of approximately 25 mm, and (c) after propagating through the Schwarzschild with a shorter distance between focus and grid of 17 mm. The shorter the distance between focus and grid was, the greater the magnification due to the larger curvature of the wavefront. As a result, a reference file from a well-defined spherical wave did not fit the grid after the light propagates through the Schwarzschild, and this magnified its image. The reference image was a far-field measurement, while the highly magnified measurements with the SO were performed in the near field. In order to understand the magnification of the spot patterns from the SO, and compare the measurements with the reference pattern, a signal-processing routine was applied. This routine was based on the fact that the real pitch between holes was known. Once magnification was determined, i.e., 3×, 5×, …, the spot patterns of the SO measurements were demagnified to the scale of the reference measurement. With appropriate demagnification, the size of the Hartman patterns from the SO and the reference file could be matched. Theoretically, demagnification is an analytical subtraction of the defocus term. The demagnified pattern, therefore, had the same pitch as that of the reference file, and could be analyzed with the centroid method. Our analysis was valid when the defocus coefficient was considerably larger than the other coefficients. Therefore, the wavefront can be seen as a quasiplanar wavefront with a spherical term.

For the wavefront evaluation, two approaches were compared: the classical analysis of the centroids and the Fourier demodulation technique. A sketch of both methods is shown in Figure 3. The former measures the relative position change of the geometrical center of intensity for each pinhole in the Hartmann array in order to obtain an estimate of the local slope of the wavefront. The technique is well-known and has been used in a large number of laboratories for real-time beam characterization, laser-aberration description, and optics alignment [25,26,27]. The centroid method was specifically carried out with Imagine Optic software WaveView.

Due to the strong curvature of the phase during the measurement of the focused beam causing magnification of the grid pattern, special treatment of the data was employed (as described above). Spots created on the camera were found far from the detection grid used in Imagine Optic’s algorithm for NA < 0.15. The SO divergence overcame this limit (NA = 0.19); therefore, in the reconstruction algorithm, the spot pitch was set as 3 times the hole pitch to correlate beamlets and centroid areas.

The second method, the concept of Fourier demodulation (FD), was adapted from atmospheric science. In this field, the wavefront sensor is used to adjust adaptive optics designed to discern astronomical objects that are distorted by the atmosphere due to a different refraction index induced by thermal changes [17,28]. Wavefront deformation for each pinhole can be defined as a change in the modulation of the entire diffraction pattern. In other words, FD is based on the fact that an aberrated wavefront modifies the periodicity of the grid; therefore, the wavefront can be calculated through Fourier transformation. Phase gradients are extracted from the Fourier transformation of the diffraction pattern. Information about the phase gradient is encoded in the first side lobes in Fourier space. The algorithm has four stages: (i) Fourier transforming the measured intensity pattern, (ii) isolating the first vertical and horizontal side lobes, (iii) centering the lobe, and (iv) transforming it back to receive the corresponding slope components. Hilbert transformation with Tikhonov’s regulation is used to integrate the extracted slopes [29,30]. The Fourier demodulation routine was developed at DESY on the basis of the Ribak et al. formulation [31].

In order to compare both methods, we added a set of orthogonal functions over the whole aperture. The sum of these orthogonal modes corresponds to the Zernike polynomials. Table 1 shows the Zernike aberration functions used in this paper.

The reliability of the centroid method (CM) and the inhouse FD script was validated using a simulated wavefront reconstruction. The test wavefront is shown in Figure 4, which contained a strong but not pure astigmatism degree. The table next to Figure 4 contains the corresponding Zernike coefficients for the simulated wavefront. The third and the fifth columns show the coefficients obtained through FD and the centroid method. Errors given for each aberration are shown on the right column next to each method. The FD-reconstructed wavefront took a numerical error of approximately 12%. On the other hand, the centroid method showed artificial numerical errors for high orders ($Z_{\left(3,1\right)}$, $Z_{\left(3,-1\right)}$). We considered these to be insensitive errors for reconstruction since the Zernike coefficients of the test wavefront were insignificant. In general, numerical errors can be attributed to the phase-unwrapping routine used within the algorithm. Wavefront-reconstruction methods are naturally very sensitive to the pupil definition, i.e., the size and position of each code, since phase slopes are recovered within that region, and Zernike coefficients are typically used for circular pupils [32]. Here, we call “the pupil” the circle that best covered the outer boundaries of the evaluated intensity pattern. Zernike polynomials are in principle only suitable for circular apertures; therefore, the pupil was the largest circle for which the measured phase difference was evaluated.

Figure 5 shows four cases of direct-beam wavefront recovery where similar circle areas were used, but a vertical shift moved the center of mass and therefore created new boundaries. Figure 5a shows the wavefront map reconstructed with a vertical shift of −10 pixels. Figure 5b shows the wavefront map of the same measurement, but the correctly reconstructed beam. Figure 5c shows the wavefront map reconstruction with a pupil defined with a shift of +10 pixels to an upper position. Central contribution was degraded, and the peak-to-valley phase became more abrupt. Figure 5d displays the wavefront with a +30 pixel shift, which corresponded to 5% of the pupil diameter.

In FD, the Hilbert transformation used to integrate the phase slopes is inherently responsive to those given boundary values of the pupil and may cause numerical error within the integration process. In order to define the correct pupil, the FD code included steps to find the largest circle covering the padded area that had a base at the center of mass. A change in this calculation step was manually realized and it affected the integration of the slopes. Thus, a change in the boundary results in a different solution of the given differential equation.

In WaveView, the centroid of the spot image is determined by integrating in the x and y directions. Peak positions were determined up to a certain percentage of the maximal peak intensity in each 1D distribution. From the number, position, and mean distance of the peaks, the grid size, i.e., the number of grid cells in each direction, was calculated, and all grid cells with intensity greater than 0 were active to form the pupil; otherwise, the slope for that cell was not calculated. In order to obtain the Zernike coefficient from the pupil dataset, the algorithm determines the smallest round disk that overlaps all illuminated cells. This disk is then used as support for the Zernike projection. Direct Zernike derivative projection on the measured local slopes was used to determine the Zernike coefficients, and the wavefront map was calculated from this set of coefficients.

Thus, successful reconstruction of a wavefront with both methods relies on accurate definition of the pupil and a numerically stable phase-unwrapping algorithm. When the given conditions were met, the FD and centroid methods were benchmarked against the validation wavefront. A comparison of the FD and CM wavefront maps provides results that especially match qualitatively, since both reconstructions demonstrated robust recovery of the main aberration.

## 3. Results

### 3.1. Direct Beam Wavefront Analysis

Figure 6 on the left shows intensity and wavefront analysis calculated using WaveView for the direct beam. On the right, the figure presents intensity and wavefront analysis determined by Fourier demodulation for the same measurements. As observed in Figure 6a,b, intensity was well-distributed for direct-beam illumination. The FEL was radiating single-mode pulses with Gaussian-like distribution. The first four terms (piston, tilts, and defocus) were subtracted. In total, we considered 12 Zernike coefficients.

Figure 7 compares individual aberrations obtained by each method. The FD method was in general agreement with the centroid method, although discrepancy was found in the vertical coma and the primary spherical. The difference for these cases was less than 1 λ for the coma and smaller than 2 λ for the spherical aberration. The peak to valley of both figures had negligible difference, this being ±4.6 λ. The reconstructions of the intensity trace and the wavefronts were alike in shape. As mentioned before, the difference may have been caused by the definition of the pupil used in each method.

The largest aberration contribution is astigmatism. Former studies at FLASH [10,33] showed that FEL pulses contain residual aberrations. The astigmatism was possibly introduced by a combination of intrinsic aberrations from the generation of the FEL beam and distortions induced by the four plane beamline mirrors (two in the tunnel and two in the experiment hall). The plane mirrors directed the FEL beam from the undulator source to the beamline end of beamline FL21. WavePropaGator [34] was used to simulate the wavefront propagation of the beam (Figure 8); the simulation was highly precise. The residual slopes and the height profile of the mirrors were all measured at the Helmholtz Zentrum Berlin by F. Siewert et al. before installing them in the FLASH beamlines [35,36]. The profiles of the two plane mirrors in the tunnel were included in the wavefront simulation, as well as the correct distances between optical elements along the beamline. Calculations confirmed that small beam distortions that originated from the two plane beamline mirrors caused an elliptical beam size at the end of the beamline of 10.9 × 10.7 mm (4-sigma).

### 3.2. Schwarzschild Aberration Analysis

Figure 9 shows the intensity and wavefront maps analyzed by the centroid method and Fourier demodulation. Figure 9a shows obtained results using the WaveView software, and Figure 9b shows results from the FD method. The projection of the SO on the sensor was altered by changing the alignment between the convex and concave mirrors of the SO. The clear aperture of the beam was an ellipse of approximately 21 × 25 mm^2^. The pattern obtained a magnification of approximately 3×. In the intensity maps, the intensity’s center of mass was shifted slightly towards the right. Figure 9c shows the obtained wavefront map after simulating and performing misalignments of our setup in optical-design software Zemax [37]. A radial shift of about 1 mm between the centers of curvature of the convex and concave mirrors explained the large contribution of the vertical coma, shown in both CM and FD. This effect is shown in the green area at the bottom-right corner of the Schwarzschild wavefront map.

Again, disagreement between both wavefront maps, including with regard to peak to valley, may have originated from the selection of the pupil, as in the direct-beam case. When the pupil is not a circle, but rather a ring, the definition of the pupil becomes more complicated. Despite that, a circular and an annular pupil provide similar Zernike coefficients when the internal ring of the annular pupil becomes smaller, and its diameter tends to zero. Mahajan et al., and Guang-ming developed an analytical solution for those cases where central shadowing is pronounced [38,39]. In our specific case, during SO wavefront analysis, the Zernike polynomial for the vertical coma was approximately 0.28 μm when analysis was realized considering a circular pupil, and shrank to 0.16 μm for annular assumption.

### 3.3. Schwarzschild Objective Focus

A potential of Schwarzschild objectives is its small focus. EUV photolithography uses microscopy as the condenser optic to focus radiation onto the mask. Zemax simulations calculated a 12 μm full width at half maximum (FWHM) focus of our SO. Figure 10 shows 80 μm propagation of FEL rays in steps of 20 μm. The coma aberration was noticeable, and good alignment of the Schwarzschild would provide even a smaller focus. Nonetheless, the focus size obtained with simulations could be still considered as ideal since the simulation considered a point source and did not include the distortion that the mirror surface may have induced, or the diffraction effects at the border of the mirrors.

The spot diagram supported the results obtained with the centroid and Fourier demodulation methods. However, to verify both techniques, the real dimensions of the focal spot could be measured in a future experiment using poly(methyl methacrylate) (PMMA) imprints [40,41]. Scanning wavefront maps of the Schwarzschild objective with minimal spherical aberration led to the smallest focus. Printing patterns on well-characterized resistance materials could demonstrate the potential of a well-aligned SO for EUV lithography.

## 4. Conclusions

We characterized a 0.19 NA Schwarzschild objective illuminated with FEL single-shot pulses using a high-numerical EUV Hartmann wavefront sensor. The experiment was performed at the open-port beamline FL21 at FLASH2. Testing the ability of the new invacuum HWS to measure large incident angles was a key point in carrying out this experiment. The HNA HWS could be fully placed into the vacuum to increase proximity to the studied optics, and grid parameters were designed to precisely study optics in the extreme ultraviolet regime with NA > 0.1. On the basis of this proof-of-principle characterization, an invacuum at-wavelength alignment of high-numerical-aperture optics such as the Schwarzschild objective is feasible.

Wavefront analysis was conducted using two methods: WaveView software, which is based on the classical centroid algorithm, and the Fourier demodulation method. Both techniques yielded comparable results, and the reliability of both methods was demonstrated by comparing them with a well-known simulated wavefront. Fine-tuning the boundary definition of the pupil plays an important role in the accuracy of the wavefront reconstruction, and explains the disagreements. Due to the sensibility of the methods at this point, qualitative comparison of the wavefront maps was more appropriate. Measurements showed good agreement with optical simulations using Zemax, which allowed for the reconstruction of misalignments.

A simulated propagation spot diagram provided a minimal focal spot of the SO of approximately 12 μm diameter FWHM. The Schwarzschild objective had very sensitive alignment, and the misalignment between the two mirrors of the system produced a strong coma that affected the dimensions of the focus. However, at-wavelength measurements provided better focal-spot control and characterization than prealignment using only visible light does. Therefore, further measurements based on PMMA imprints using strong focusing optics, i.e., the Schwarzschild objective or off-axis parabola mirrors, approach the ground truth of wavefront analysis presented in this work.

This proof-of-principle experiment opens up the possibility for the characterization and suppression of aberrations for high-numerical-aperture microscopes by Hartmann wavefront sensing, and thus their use in EUV lithography.

## Figures and Tables

**Figure 1 sensors-20-06426-f001:**
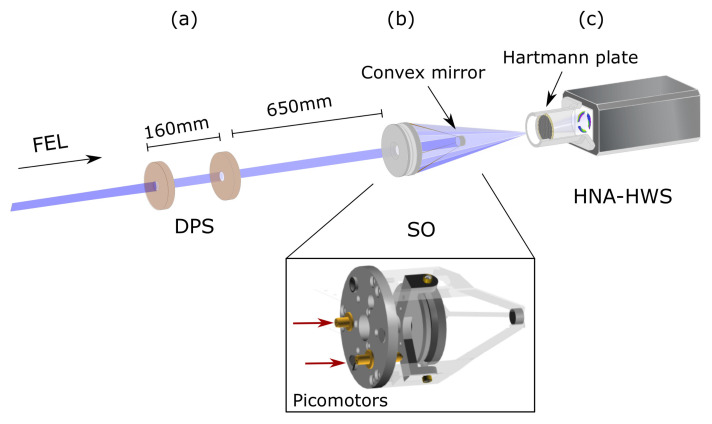
Experiment setup at FLASH. (**a**) Radiation clipped by aperture set (differential pumping stage, DPS) in front of Schwarzschild objective. (**b**) Schwarzschild optics reflects the radiation from FEL and (**c**) focuses it upstream of the Hartmann plate of high-numerical-aperture Hartmann wavefront sensor (HNA-HWS).

**Figure 2 sensors-20-06426-f002:**
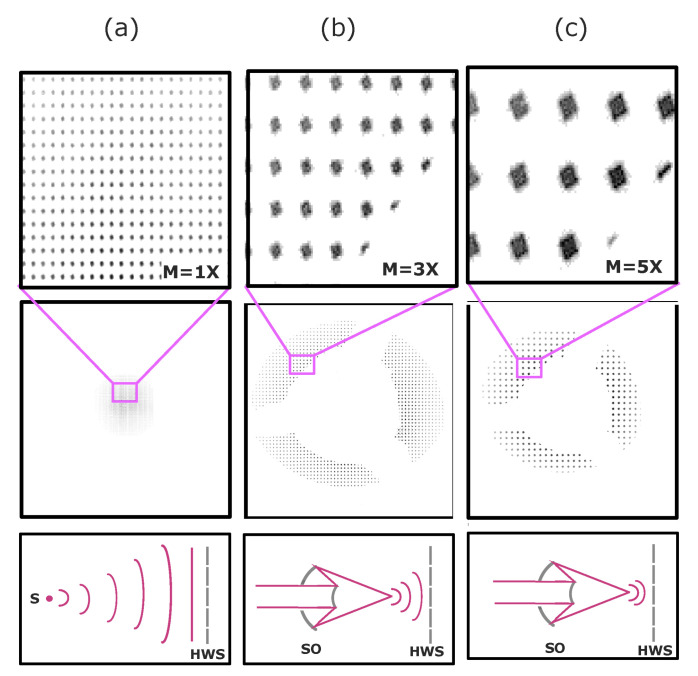
Hartmann pattern overview of direct beam compared with pattern on the HNA-HWS after it propagated through the Schwarzschild at different distances. (**a**) Beam propagated directly from the source (S) and did not magnify the Hartmann plate. (**b**) Pattern through Schwarzschild objective (SO) when it was focused 25 mm apart from the grid, introducing M = 3× magnification due to wavefront curvature. Annular pattern was divided in three sections as a consequence of the three fins of the mount that supported the concave mirror. For an SO focusing closer, i.e., 17 mm from the grid (**c**), the wavefront curvature was stronger, the magnification effect rose to M = 5×, and total pattern dimensions were decreased.

**Figure 3 sensors-20-06426-f003:**
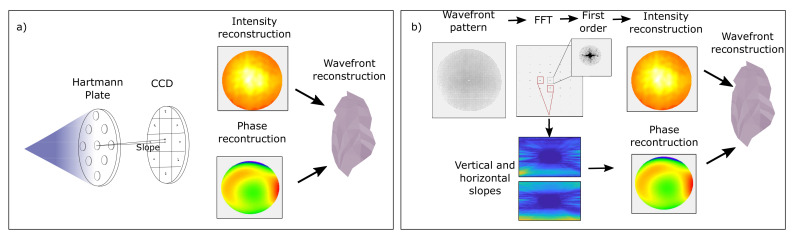
(**a**) Classical centroid method based on measuring relative position change of geometrical center of intensity of each pinhole in Hartmann array to obtain an estimate of the local wavefront slope. (**b**) Fourier demodulation based on the fact that an aberrated wavefront modifies grid periodicity; therefore, the wavefront can be calculated through Fourier transformation. Algorithm has four stages: (i) Fourier transforming measured intensity pattern, (ii) isolating first vertical and horizontal side lobes, (iii) centering lobe, and (iv) transforming it back to receive corresponding slope components.

**Figure 4 sensors-20-06426-f004:**
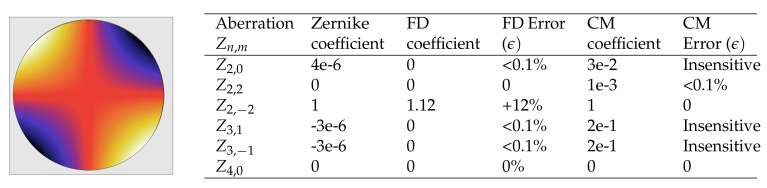
(**left**) Simulated wavefront map with strong astigmatism at 45°; (**right**) table with corresponding Zernike coefficients of simulated test-wavefront map and obtained values using our Fourier demodulation (FD) script and the centroid method (CM). Units were arbitrary and normalized to simulated oblique astigmatism. Next to the Zernike coefficients of each method, the error is shown. In most cases, this error was insensitive or <0.1%.

**Figure 5 sensors-20-06426-f005:**
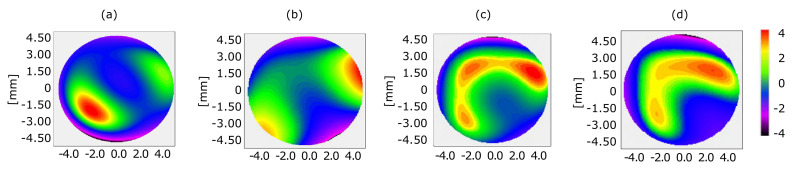
Wavefront map obtained with Fourier demodulation on direct beam by vertically displacing circular area of interest by (**a**) −10 pixels or −0.14 mm, (**b**) 0 mm, (**c**) 10 pixel or 0.14 mm, and (**d**) 30 pixels or 0.39 mm.

**Figure 6 sensors-20-06426-f006:**
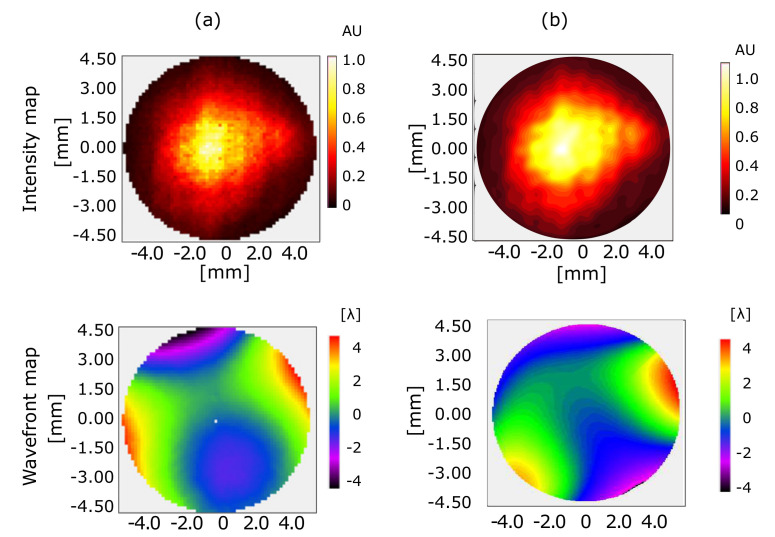
Intensity and wavefront map of direct beam obtained with (**a**) centroid and (**b**) Fourier demodulation reconstruction methods.

**Figure 7 sensors-20-06426-f007:**
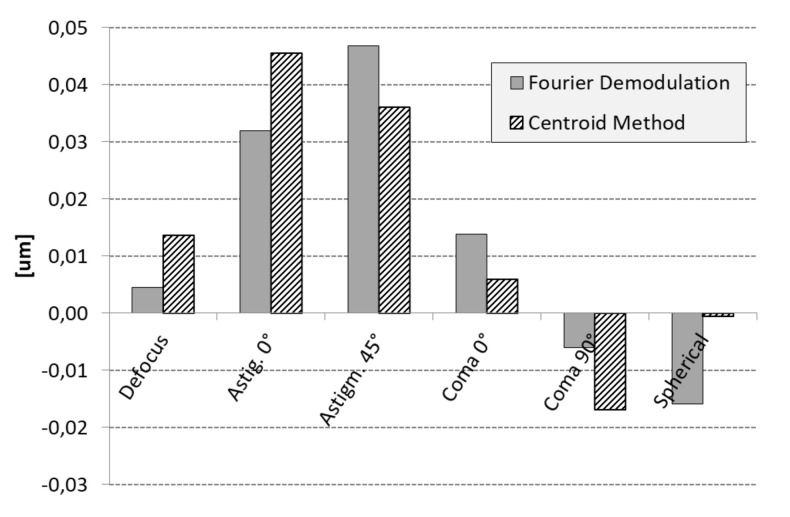
Calculated Zernike coefficients using Fourier demodulation (gray bars) and centroid method (bars with black lines) for the direct beam. Major distortions were found for vertical and horizontal astigmatism.

**Figure 8 sensors-20-06426-f008:**
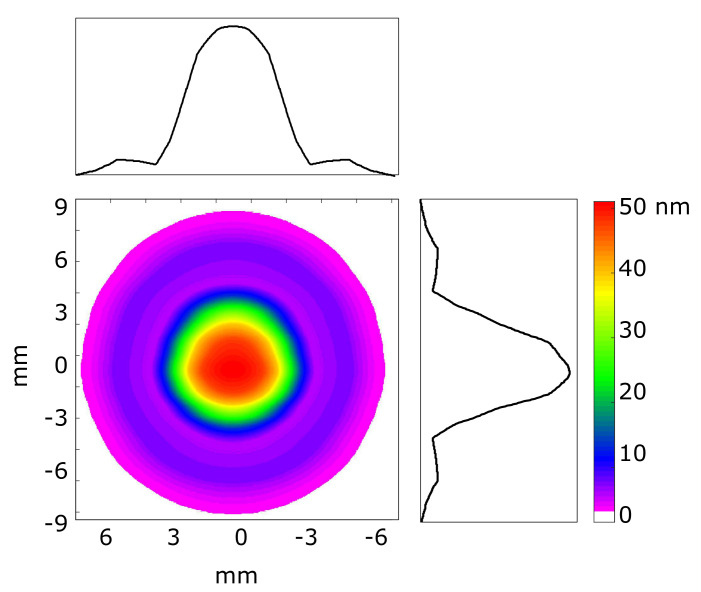
Wavefront simulation of beam phase at the end of beamline FL21. Beam only contained distortions from the two plane beamline-mirror profiles. Perfect Gaussian source was assumed to start propagation.

**Figure 9 sensors-20-06426-f009:**
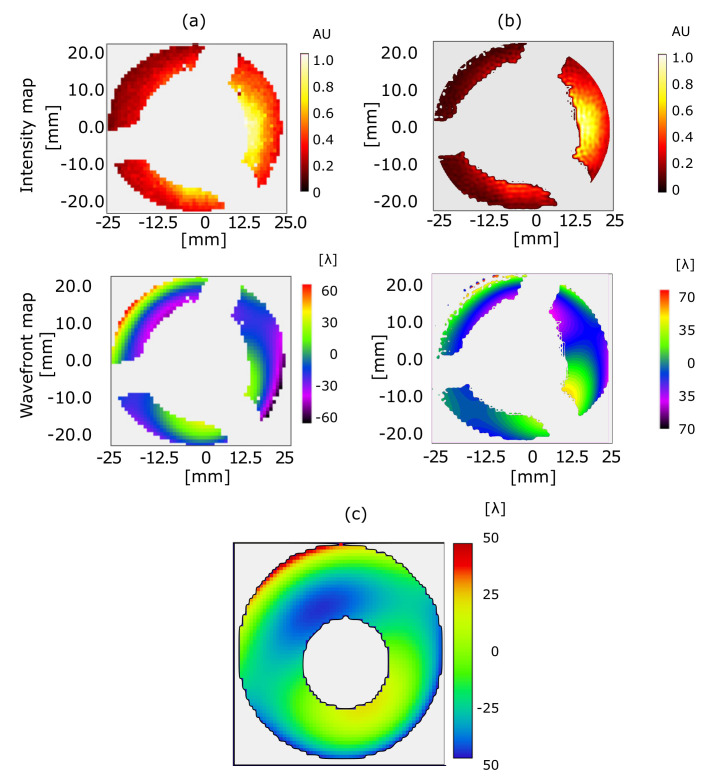
Intensity and wavefront map of Schwarzschild pattern obtained with (**a**) centroid and (**b**) Fourier Demodulation reconstruction methods. (**c**) Zemax simulation of the wavefront map.

**Figure 10 sensors-20-06426-f010:**
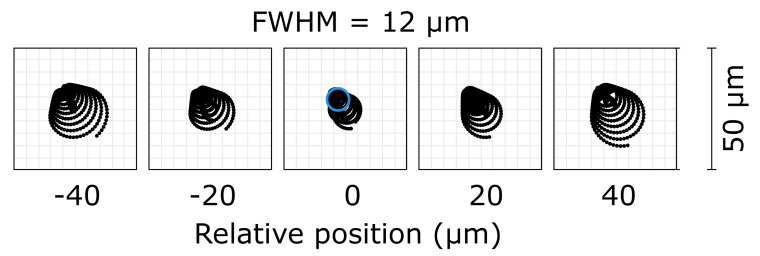
Footprints around focal spot of Schwarzschild in steps of 20 mm. Full width at half maximum (FWHM) of the central one corresponded to 12 mm.

**Table 1 sensors-20-06426-t001:** Zernike polynomials with each mode representing type of aberration that is described by a polynomial. All polynomials depend on two parameters, azimuthal angle ϕ and radial distance ρ.

Aberration	n,m	Function
Defocus	$Z_{2,0}$	$2\rho^{2}-1$
Astigmatism horizontal	$Z_{2,-2}$	$\rho^{2}\cdot cos\phi$
Astigmatism oblique	$Z_{2,2}$	$\rho^{2}\cdot sin\phi$
Coma horizontal	$Z_{3,-1}$	$3\rho^{3}-2\rho \cdot cos\phi$
Coma vertical	$Z_{3,1}$	$3\rho^{3}-2\rho \cdot sin\phi$
Spherical	$Z_{4,0}$	$6\rho^{4}-2\rho -1$

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
