# Peer review of "Wavefront Sensing for Evaluation of Extreme Ultraviolet Microscopy"

_sensors, 2020, doi:10.3390/s20226426_

Round 1

Author Response

Dear reviewer,

The authors appreciate the insightful comments on revising all the aspects of the paper. We considered all the comments and we believe that after this review, the paper improved significantly. We added some pictures and extended the explanations, overall the theoretical part. To help to organize better the manuscript we added subsections. The new paragraphs and the specific answer to your questions and comments are in the pdf attached. We hope you considered upgraded now to understand better the study.

Reviewer 2 Report

My primary problem with the paper is that I am unable to grasp what point the authors are trying to get across here.

The challenge seems to be to gat HNA-HWS to work in order to assist aligning an SO setup.  In the conclusion, the authors claim that while the actual alignment was not possible, a proof of principle has been made thet the HWS is actually capable of supporting the task.

However, for me this claim is not supported by the presented experiments.

Apart from the description of the various steps being unclear and potentially incomplete, I cannot find any comparison to a known ground truth in the paper. The authors evaluate the HWS signal with two different methods and claim that, since both agree reasonably well there must be some truth in the result. A comparison to the outcome of an optical simulation using Zemax appears incomplete and, at least to me cannot really support that claim.  Simply applying a misalignment producing a coma aberration and then showing the propagation through the focal plane says nothing about whether the WFS signal is right or wrong.

At the very least, the Zemax simulation should include a couple of HWS sub-apertures (holes) to check whether it can reproduce the spot pattern observed. Better would be a full simulation using fourier optics to verify if the two methods described work on simulated data - with known ground truth! - as expected.  If that were granted,  the lab application would be much more credible.  Better still, albeit probably difficult to achieve, would be to analyze the real SO setup independently and find the actual aberrations.

Apart from these fundamental difficulties which I'm afraid oblige me to demand a substantial revision of the paper, I have the following comments:

P1, L24: at wavelength with... I guess there's the actual "of xx nm" missing

P1, bottom: Not being from the UV field, could you insert one sentence about why it is such a challenge to operate a CCD in vaccum?

P1, L39-40, The sentence i slightly weird as it switches from from the genral introduction to the outline of the paper without warning - maybe "In this paper, we present..."

P2,L54 I guess it's not really 10⁶ mbar in your "vacuum" chamber... add "-"

P5, L133 ff. "General agreement". Would you maybe have a comment as to why the FD method shows about 20% larger amplitudes of aberration wrt. centroid method in Fig4?  Could this be related to the discussion on P.6? If yes, maybe give a forward reference.  Additionally, in Fig5 the centroid method is detecting a large coma aberration, which the FD essentially fails to see at all.  So the sum should yield larger aberrations for CM, vice versa with respect to Fig.4. How can this be explained? Anyway, I wouldn't call this "general agreement" if one method is blind to Coma, the other is not. 

P5 L 141: "distortions are originated due to" -> "distortions originate in"

P5 L144: I'm guessing that in this line you switched from describing the direct beam to the SO setup. I t would be nice to be sure about that by having a small remark, or even a subsection headline.

Fig5: As throughout the paper, it's difficult to judge to which setup the presented results belongs. Put "direct beam setup" somewhere in the caption.

P6, L148ff: You are discussing individual aberration modes - pls. show the expnasion analogous to Fig 5, so that the reader can follow.

The comparison to the Zemax simulation is pointless, unless there's a way to prove that they indeed reproduce what is seen in the experiment. Either use Zemax to reproduce Fig 7, or find a way to directly measure Fig 8. and compare.

P6, L158: You rightly point out that Zernikes are suitable only for circular apertures, but other sets do exist: What about Annular Zernike modes or Karhunen–Loève functions? They are widely used in Astronomy for telescope apertures with secondary mirror obscurations.

P6, L161 Fig 9 is actually for which data set - direct beam or SO?

I do have a question on the "calibration" of the HWS: When the HASO software evaluates the controids, what are the "zero" centroid positions that you assign to a "flat" wavefront?  I found no mention of that in the paper, and I guess it's difficult to actually measure them since the flat wavefront exists only (at least approximately) in the direct setup and produces no spots at the locations where spots are found in the SO setup. Are the zero positions determined by simulation?  In other words: The aberrations you determine are relative to what??? Without that knowledge, it's actually even more difficult to judge whether the method can help aligning something or not.

Author Response

The challenge of the paper is to demonstrate that large numerical aperture optics i.e. Schwarzschild objective, can be evaluated using wavefront sensing. We believe that the new pictures, extended explanations help the script to support better the task. Moreover, new subsections have been included in the manuscript to organize better the text. This was the first measurement on optics ever done with a high numerical aperture Hartmann wavefront sensor. Indeed, the measurements were just the first steps to develop a new method for the wavefront analysis of high numerical aperture optics. It was very useful to determine the tools needed in the future to actually align a Schwarzschild objective.

The extended review and effort done by the referee was immensely helpful, and we, the authors appreciate the insightful comments on revising all the aspects of the paper. We considered all the comments and we believe that after this review, the paper improved significantly. All the changes to support the improvement, as well as specific answers to the reviewer are in the attached document.

Round 2

Reviewer 2 Report

First of all let me thank you for the prompt and extensive revision of the manuscript.  I can see the point now.

For me two questions remain, though:

  The top half of page 4 describes how the wavefront measurement works.  I think I can guess now that the reference positions of the spot grid are simply the positions of the holes in the Hartmann plate.  This is of course different from what I'm used to, where instead of a hole plate a lenslet array is used and the zero-positions of the spots need to be calibrated by illuminating the setup with a flat wavefront. Hence my initial difficulties in understanding the procedure.  Nevertheless, the software needs to relate the individual spots to individual zero positions, and due to the SO, the pattern is totally different from a flat-wavefront one.  How is the relation of a particular spot with its specific zero-position achieved?  Maybe all the information is in the HASO manual, but a tiny little bit of more information would I think be beneficial to the paper.

  Second, the question of ground truth: there is still no independent verification of the result.  You're using different methods, but they rely on the same measurement, in fact the same data.  I understand now that your prime intention is a proof of principle, but maybe you could include a small outlook of how your finding could potentially be verified in a future experiment? Maybe at least by verifying Fig. 10 in an experiment? 

In any case I consider these questions interesting and leave it to your discretion to address them or not.  The paper could also be published as is - after all the missing cross references and citations have been fixed!

Author Response

Foremost, the author are very glad about the positive feedback of the first revision. Indeed we found the next two questions very interesting. We address them in the following lines and we really hope these answers meet with the reviewer approval.

First of all let me thank you for the prompt and extensive revision of the manuscript.  I can see the point now.

For me two questions remain, though:

  The top half of page 4 describes how the wavefront measurement works.  I think I can guess now that the reference positions of the spot grid are simply the positions of the holes in the Hartmann plate.  This is of course different from what I'm used to, where instead of a hole plate a lenslet array is used and the zero-positions of the spots need to be calibrated by illuminating the setup with a flat wavefront. Hence my initial difficulties in understanding the procedure.  Nevertheless, the software needs to relate the individual spots to individual zero positions, and due to the SO, the pattern is totally different from a flat-wavefront one.  How is the relation of a particular spot with its specific zero-position achieved?  Maybe all the information is in the HASO manual, but a tiny little bit of more information would I think be beneficial to the paper.

The information is unfortunately not available in the HASO manual since the Hartmann wavefront measurements on a Schwarzschild have been done for the very first time in this experiment. For data analysis, we have scripted a preprocessing (image analysis) tool. Its essence has been explained in the following lines: 115- 127.

 The large beam divergence produces a magnified diffraction pattern at the camera. The standard algorithm could be used, as soon as a de-magnification of 3X was applied to the measured diffraction patterns. Now the text reads as:

“In order to understand the magnification of the spot patterns from the SO and compare the measurements to the reference pattern, a signal processing routine was applied. This routine is based on the fact that the real pitch between holes is known. Once the magnification is determined, i.e., 3X, 5X …, the spot patterns of the SO measurements were de-magnified to the scale of the reference measurement.”

  Second, the question of ground truth: there is still no independent verification of the result.  You're using different methods, but they rely on the same measurement, in fact the same data.  I understand now that your prime intention is a proof of principle, but maybe you could include a small outlook of how your finding could potentially be verified in a future experiment? Maybe at least by verifying Fig. 10 in an experiment? 

We found very interesting the proposal of the reviewer of including some outlook about a future experiment. Fig. 10 can indeed be also measured using PMMA imprints. The future experiment will be develop as follows: 1) scanning wavefront maps using the wavefront analysis shown in this paper, 2) Back propagate the beam for each scan, 3) imprints the focus on a PMMA and check if the aberrations match the prints.

Section 10 (Lines 258-263) reads now as:  “The spot diagram supported the results obtained with the centroid and the Fourier Demodulation method. However, to verify both techniques the real dimensions of the focal spot could be measured in a future experiment using PMMA imprints [40,41]. Scanning wavefronts maps of the Schwarzschild objective with minimum spherical aberration lead to the smallest focus. Printing patterns on well-characterized resist materials will demonstrate the potential of a well-aligned SO for EUV lithography.”

In any case I consider these questions interesting and leave it to your discretion to address them or not.  The paper could also be published as is - after all the missing cross references and citations have been fixed!